# Lateral pressure equalisation as a principle for designing support surfaces to prevent deep tissue pressure ulcers

Colin J. Boyle[1,2]*, Diagarajen Carpanen[1], Thanyani Pandelani[1], Claire A. Higgins[1], Marc A. Masen[2], Spyros D. Masouros[1]

1 Department of Bioengineering, Imperial College London, London, United Kingdom, 2 Department of Mechanical Engineering, Imperial College London, London, United Kingdom

* c.boyle@imperial.ac.uk

**Data Availability Statement:** All data required to reproduce the models in the paper are available from figshare using DOI: 10.6084/m9.figshare. 10510787.

## Abstract

When immobile or neuropathic patients are supported by beds or chairs, their soft tissues undergo deformations that can cause pressure ulcers. Current support surfaces that redistribute under-body pressures at vulnerable body sites have not succeeded in reducing pressure ulcer prevalence. Here we show that adding a supporting lateral pressure can counteract the deformations induced by under-body pressure, and that this 'pressure equalisation' approach is a more effective way to reduce ulcer-inducing deformations than current approaches based on redistributing under-body pressure. A finite element model of the seated pelvis predicts that applying a lateral pressure to the soft tissue reduces peak von Mises stress in the deep tissue by a factor of 2.4 relative to a standard cushion (from 113 kPa to 47 kPa)—a greater effect than that achieved by using a more conformable cushion, which reduced von Mises stress to 75 kPa. Combining both a conformable cushion and lateral pressure reduced peak von Mises stresses to 25 kPa. The ratio of peak lateral pressure to peak under-body pressure was shown to regulate deep tissue stress better than under-body pressure alone. By optimising the magnitude and position of lateral pressure, tissue deformations can be reduced to that induced when suspended in a fluid. Our results explain the lack of efficacy in current support surfaces and suggest a new approach to designing and evaluating support surfaces: ensuring sufficient lateral pressure is applied to counteract under-body pressure.

## 1. Introduction

Supporting the body weight of critically ill, immobilised or paraplegic people without causing soft-tissue injury is not an easy task. The loading induced while lying or sitting for prolonged periods can cause damage to skin, adipose tissue and muscle; this damage is known as a pressure ulcer. Pressure ulcers are estimated to affect one in five hospitalised patients in Europe [1], while prevalence in some patient groups are much higher. For example, 85% of spinal cord injury patients develop a pressure ulcer over their lifetime [2] with associated care costs of

**Funding:** This project was partly funded by an Imperial College London and Imperial Innovations Ltd. Proof of Concept funding to CB and SM and EPSRC funding to CH, MM and SM (EP/N026845/1). The EPSRC had no role in study design, data collection and analysis, decision to publish, or preparation of the manuscript. Imperial College London provided support in the form of salaries for authors [CJB, SDM], but did not have any additional role in the study design, data collection and analysis, decision to publish, or preparation of the manuscript. To clarify further, Imperial Innovations Ltd. is a technology-transfer company that acted as the tech-transfer office of Imperial College London from 1986 to 2019. The awarding of Proof of Concept funding or any other associate with Imperial Innovations Ltd does not alter our adherence to PLOS ONE policies on sharing data and materials.

**Competing interests:** I have read the journal's policy and the authors of this manuscript have the following competing interests: CB and SM have filed a patent (application number 1814813.0) for a support surface device based on the principle described in this paper. Imperial Innovations Ltd and Imperial College London partly funded this work through a Proof of Concept grant to SDM and CJB. This does not alter our adherence to PLOS ONE policies on sharing data and materials. Drawing from this work, CJB and SDM have applied for a patent (GB1814813.0 "Device for supporting a body part"). This does not alter our adherence to PLOS ONE policies on sharing data and materials.

approximately $1.2 billion annually in the US [3]. A severe form of pressure ulcer develops in subdermal tissue close to bony prominences such as the ischial tuberosity and sacrum of the pelvis [4,5], and is known as a deep tissue injury. Because of the severity of deep tissue injury, preventative strategies have been a major focus in the field.

One approach to preventing pressure ulcers is to design support surfaces to reduce pathological pressures, and this has been a major area of research for the past forty years [6]—indeed 'invalid beds' have been developed since the 19[th] century [7]. While support surfaces have become increasingly high-tech, they have yet to outperform high-specification foam mattresses, and their adoption in clinics has not led to a significant reduction in pressure ulcer prevalence [8,9]. While this lack of progress may indicate that we have reached the limit of support surface design, in this paper, we argue that current designs have been based upon a suboptimal design principle—that of under-body pressure re-distribution.

The presumption that high surface pressure leads to pressure ulcers, and therefore should be reduced, seems obvious. However, experimental, computational and clinical evidence suggests that high surface pressures do not necessarily cause pressure ulcers. Peak surface pressures (as measured by pressure mapping sensor arrays) cannot identify at-risk patients [10,11]. High-tech mattresses that reduce peak surface pressures have increasingly been adopted in clinical settings yet their impact on pressure ulcer prevalence has been disappointing [8,9]. Furthermore, soft tissue *can* tolerate extremely high surface pressures under certain circumstances. The soft tissues of a deep-sea diver, for example, are exposed to 100 kPa of surface pressure for every 10 m descended, yet pressure-related injuries to soft tissues are not a common issue in diving [12,13]. Computational studies have helped to explain these observations, with Oomens et al. [14] demonstrating that peak surface pressure has very little impact on internal deformations near bony prominences—regions where deep tissue injuries are likely to occur [14,15]. Since reducing peak surface pressure has failed to protect deep tissue, we sought to determine if there is any way to manipulate the external pressure profile that can protect deep tissue.

Deep tissue pressure ulcers develop as a result of several overlapping processes: ischaemia [16], ischaemia-reperfusion injury [17] lymphatic network obstruction [18,19] and direct cell deformation [20]. Each of these processes is triggered by excessive deformation (exacerbated by shear stresses, microclimate, and other risk factors) within soft tissue, and so in some regards, a pressure ulcer could be more aptly named a 'deformation ulcer'. Redistributing surface pressure (as current devices aim to do) does not necessarily reduce deformations (and hence pressure ulcers) because soft tissue has very different tolerances to the two components of stress (Fig 1A): deviatoric stress (which tends to change the shape of an object) and dilatational stress (which tends to change the volume of an object, but not its shape) [21,22]. Human soft tissues are almost incompressible [23], and so can tolerate high dilatational stress with minimal deformation. In contrast, soft tissues deform readily with deviatoric stress, therefore it is this stress that must be minimised to prevent ulceration. While submersed, a diver experiences nearly uniform pressures on all surfaces; this tends to induce dilatational stress [22]. On the other hand, interaction with very localised surface pressure–such as when sitting on a chair or lying on a mattress–induces large deviatoric stress (and therefore deformations) as the soft tissue bulges and is displaced laterally away from the load (Fig 1B).

One way to prevent excessive deformations may be to restrain the soft tissue from deforming by applying a supporting lateral pressure. In this paper, we test the plausibility of this principle using a computational model of the weight-bearing pelvis in a seated individual. We hypothesise that actively applying pressure laterally to the soft tissue of the pelvis will reduce the deformation at the ischial tuberosity to a greater extent than the commonly applied method of redistributing under-body pressure (Fig 1B). Our rationale for this work is that by

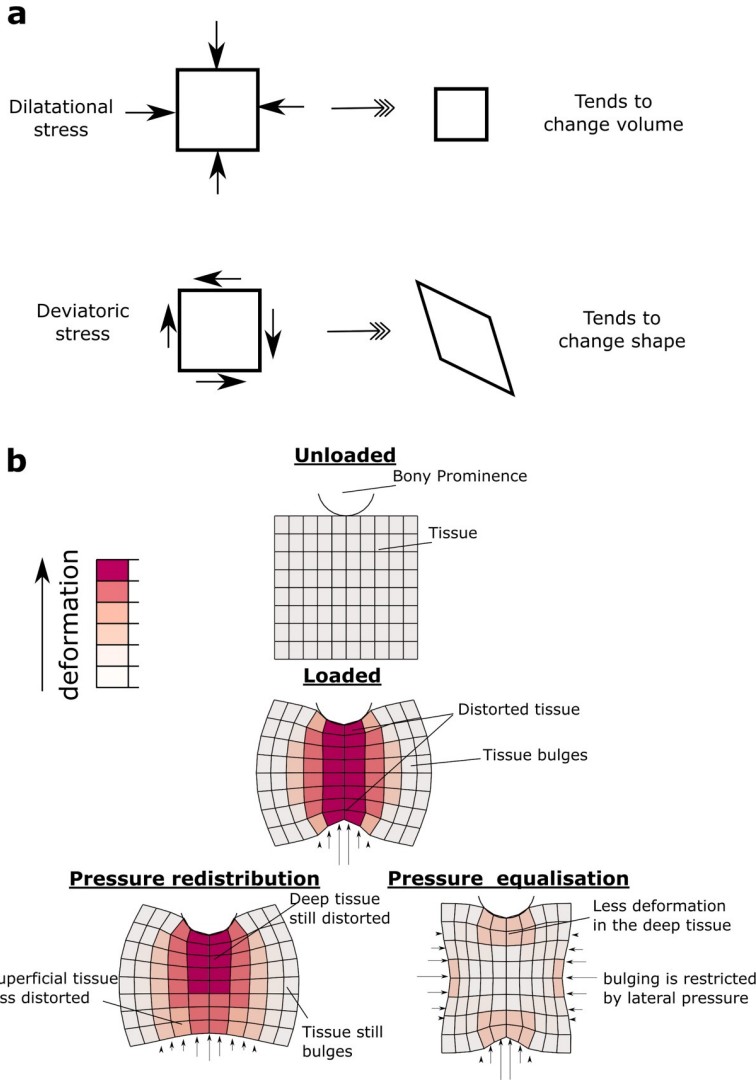

**Fig 1. Deformations beneath a bony prominence.** The stress in soft tissue has two components, dilatational and deviatoric (a). Soft tissue is much more resistant to dilatational stress than deviatoric stress. Under a bony prominence, the soft tissue is distorted due to the concentrated pressures at the bone and the support (b). Redistributing the surface pressure has some effect on the outer (superficial) region, but not on the deep tissue. We hypothesise that by applying pressure laterally (termed pressure equalisation), bulging is reduced, and the tissue can bear the load in a more dilatational mode.

making subtle changes to the design objectives used for support surfaces, we may be able to substantially reduce the risk of ulceration for high-risk patients.

## 2. Methods

First, we adapted a previously-developed finite element model of seated buttocks [14] to test the hypothesis that applying lateral pressure will reduce tissue deformations (section 2.1). The seated position was chosen because the ischial tuberosity is a common site for deep tissue injury [1]. Next, we used the model to determine whether applying lateral pressure or changing the stiffness of a standard cushion has the greatest effect on deep tissue deformations (section 2.2). To ensure that these effects translate to a more realistic setting, we developed a 3D

model from MRI scans (section 2.3). Finally, we sought to formalise the relationship between surface pressure and internal deformations into a design principle—equalising under-body pressure with lateral pressure. To do this, we described the interaction of soft tissue and a support surface as a surface pressure boundary condition, which could be manipulated and studied independently of particular cushion design (section 2.4). All finite element input files and analysis protocols are available in an open database (10.6084/m9.figshare.10510787).

## 2.1 Model of the seated pelvis with lateral pressure application

2.1.1 Geometry and material models. An axisymmetric geometry was used to model the soft tissue surrounding a single ischial tuberosity in a seated individual. The geometry was similar to that used by Oomens et al. [14] but included more of the pelvis soft tissue to allow lateral pressure to be modelled (Fig 2A). The soft tissue was partitioned into fat, muscle and skin to produce similar patterns as found from MRI imaging (Fig 2B).

There have been many material models of skeletal muscle [24,25], skin [26], and to a lesser extent fat [27]. However, experimentally-based models that quantify all three tissues together are rarer, making it difficult to combine tissues defined from different experimental setups. In this study, we used the material models based on Oomens et al [14] because it enabled direct comparison with that study, and because all three soft tissues were characterized. Each region was assigned an Ogden hyperelastic material model, and parameter values are listed in Table 1. A flat, 76 mm-thick, two-layered seat cushion was modelled with hyperelastic material properties representing a soft cushion (Table 1). An air-filled chamber (the pressure-equalisation device) was introduced to apply a lateral pressure to the soft tissues (Fig 2A). This was shaped to conform to the seated pelvis and was modelled using membrane elements that can resist tensile, but not bending, loads. The chamber wall material was modelled as sufficiently stiff (Young's Modulus, E = 10 MPa) so as not to allow appreciable changes in length.

2.1.2 Boundary conditions. The proportion of body weight borne by the ischial tuberosities while seated varies from 18% to 77% [14]. We estimated the amount of load supported by the pelvis at 400N —representing approximately 50% of the body weight of an 80 kg adult (with each tuberosity bearing 200N) because it is within the range of experimental findings and enables direct comparison with Oomens et al [14]. Symmetry boundary conditions were prescribed to all nodes lying along the z-axis (Fig 2A). The cushion base was constrained in all directions. Two nodes of the pressure equalisation device were constrained in all directions, and a uniform pressure was applied to the inner surface of the chamber to a maximum of 80 kPa. Frictionless contact was assumed between the support surfaces and the skin. While this is a simplification of the real-world scenario, a sensitivity study revealed friction to have a negligible effect on model predictions (S2 Appendix). Normal contact behaviour was enforced using the penalty method with finite-sliding [28].

2.1.3 Solution approach and output. A mesh sensitivity analysis was performed leading to a final mesh of 13056 linear quadrilateral elements representing the soft tissues. All models were solved as quasi-static, non-linear analyses using the ABAQUS finite element software (v2016, Dassault Systems, France). To analyse and compare models, the von Mises stresses and shear strains in the soft tissue regions were calculated. Von Mises stress ($q$) is a scalar representing the deviatoric part of the stress tensor ($S_{ij}$) and defined as $q = \sqrt{\frac{3}{2}S_{ij}S_{ij}}$, $S_{ij} = \sigma_{ij} + p\delta_{ij}$, $p = -\frac{1}{3}\sigma_{ii}$, where $\sigma$ is the stress tensor and $\delta_{ij}$ is the Kronecker delta function. Shear strain was calculated as $\epsilon_1 - \epsilon_3$, where $\epsilon_1$, $\epsilon_3$ are the maximum and minimum principal strains, respectively. These were chosen to represent the level of deviatoric stress and strain, respectively.

To summarise the stresses and strains in the deep tissue, we sampled 1600 elements within a 30mm radius of the ischial tuberosity. Since elements vary considerably in size throughout

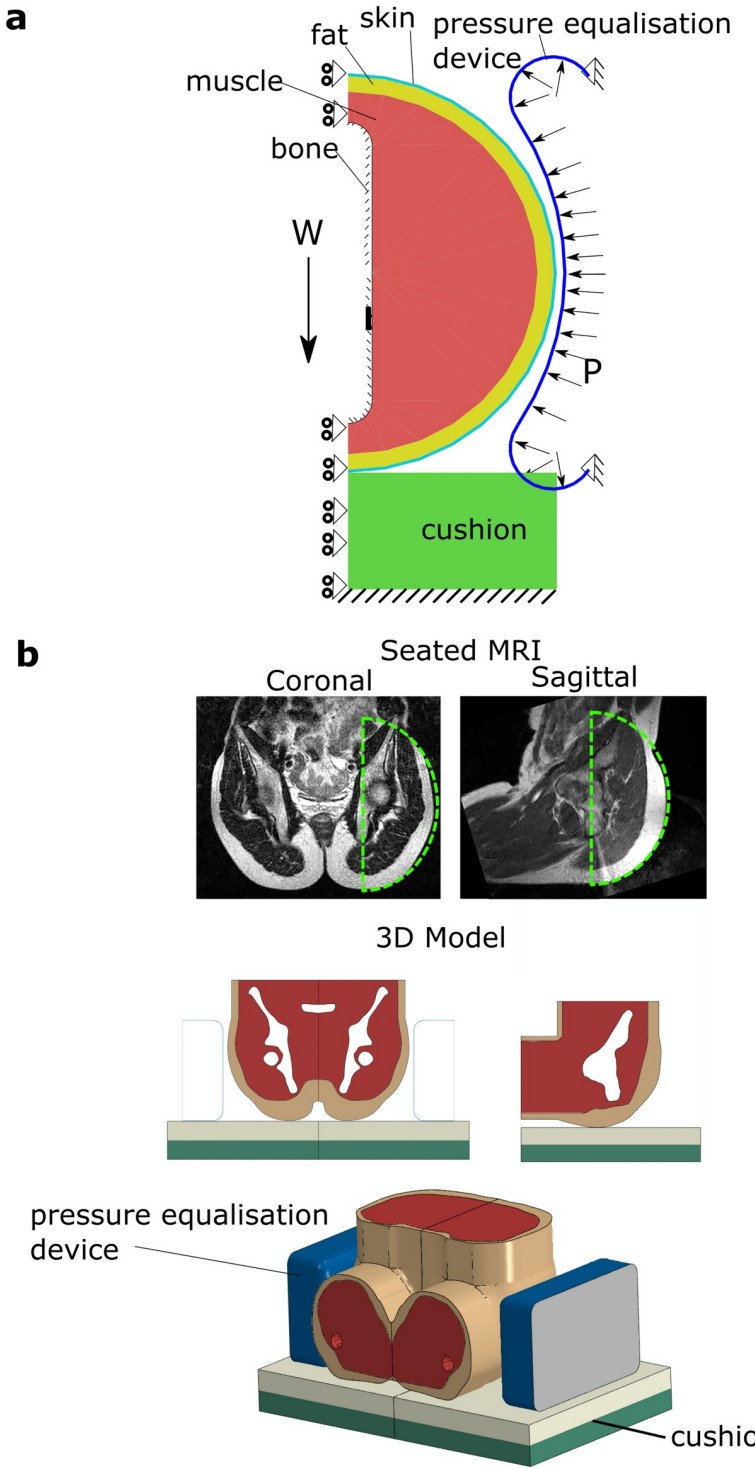

**Fig 2. Finite element models.** (a) An axisymmetric model of the soft tissue surrounding the ischial tuberosity. The model incorporates a rigid bony prominence, muscle, fat and skin layers interacting with a cushion and a pressure equalisation device. Axisymmetry was assumed, which allowed a force-controlled simulation of weight-bearing (W is the load borne by the ischial tuberosity). The pressure equalisation device was modelled as an air-filled chamber with a controllable internal pressure, P. (b) The axisymmetric region modelled is shown superimposed on saggital and coronal MR images of a seated male (top). A 3D model was generated from the MR images to assess 3D deformations (bottom).

**Table 1. Parameters for the Ogden material model for each of the materials modelled.** Ogden strain energy density function, $U = \frac{2\mu}{a^2}\left(\lambda_1^a + \lambda_2^a + \lambda_3^a - 3\right)$, where $\lambda_{1,2,3}$ are the principal stretches, $\mu$ and $a$ are material constants and $U$ is the strain energy density.

| Material | $\mu$ (MPa) | $a$ (-) |
|---|---|---|
| Skin | 0.04 | 30 |
| Fat | 0.025 | 10 |
| Muscle | 0.045 | 5 |
| Stiff cushion | 0.08 | 10 |
| Medium cushion | 0.005 | 10 |
| Soft cushion top | 0.0035 | 7 |
| Soft cushion bottom | 0.005 | 10 |

the region, we weighed our sampling by element volume (IVOL output from ABAQUS). For example, an element with a volume 2V is twice as likely to be sampled as an element with volume V. Weighting the results like this ensures that analyses are independent of mesh density, which varies throughout the model.

Peak stress was defined as the 95th percentile of the stress data to avoid extreme outliers that may be sensitive to boundary conditions. Effects sizes (in the mean and peak values) between models were estimated by calculating bootstrapped 95% confidence intervals [29]. We also analysed deep tissue stress along a path through the soft tissue directly beneath the ischial tuberosity, and along the surface of the ischial tuberosity.

## 2.2 Comparing lateral pressure application to changing cushion stiffness

Cushion stiffness is a design variable commonly used to create a more conformable cushion, and this was used as an intervention to compare to adding lateral pressure. The model described above was adapted to model load-bearing on three different cushion designs–a stiff, medium, and soft variety. The stiff and medium cushions were homogeneous, 38 mm thick cushions, while the most compliant (softest) was produced by defining a 76 mm thick two-layer cushion as in the previous section. The bottom layer of this cushion had the medium-stiffness cushion properties, and a softer material was assigned to the 38 mm top layer (Table 1). These cushions were based on those analysed by Oomens et al. [14], which were in turn calibrated to model materials commonly used in wheelchair cushions.

For each of the three cushion simulations, load-bearing to 200N was established as in section 2.1. Pressure in the pressure equalisation device was then increased incrementally to a maximum of 80 kPa. The stresses and strains induced when lateral pressure is applied were computed.

## 2.3 Assessing deformations in 3D

Modelling the pelvis using a 2D axisymmetric model as above requires simplification of the bone and soft tissue geometries. To ensure that the beneficial effect of adding lateral pressure translates to 3D environments, we developed a 3D model that can more accurately capture the geometry of a seated pelvis. The MRI data of a male subject (age 30) was used to generate the 3D geometry (Fig 2B) including skin, fat, muscle and bone. Data usage was approved by the Imperial College Research Ethics committee under ethical approval number ICHTB HTA licence: 12275 and REC Wales approval: 12/WA/0196. To aid comparison of results between models, material properties for each layer were assigned to be consistent with the 2D model. The nodes representing the outer surface of the pelvic bones were constrained to move in the z-direction. Central symmetry was assumed to reduce model size by half. The soft tissue was

meshed using 669,995 linear tetrahedral elements. A body force of 200N was applied to the bone nodes. This represents a full body weight of 80 kg, with the assumption that 50% of this travels through the pelvis of a seated individual (as was assumed with the axisymmetric model and is based on Oomens et al. [14], see section 2.1.2). Maintaining this level of loading in the 3D model aids comparison between that and the axisymmetric model. The lateral supports were modelled as air-filled cavities, with a thin outer membrane of material (as in the 2D model). To model the seated load case, the body force was ramped incrementally over the first analysis step. Next, the lateral supports were displaced towards the pelvis while the internal pressure within the support was fixed at 1 kPa. Finally, the pressure within the lateral support was increased to 10 kPa incrementally, and the von Mises stresses and maximum shear strains were calculated over ten increments. This model was solved using Abaqus/Explicit, with mass scaling applied to ensure kinetic energy was less than 1% of internal strain energy.

## 2.4 Determining the relationship between surface pressure and deep tissue mechanics

We studied how the shape and magnitude of the surface pressure distribution affected internal tissue stress to understand how these stresses can be minimised. The specific cushions and lateral support used in the previous sections were removed and replaced with a surface pressure boundary condition. For our axisymmetric model, pressure is a function of the angle from vertical, $P(\theta)$. The surface pressure was constrained to ensure that the model was in static equilibrium: the sum of the vertical forces due to surface pressure is equal to body weight ($W$), ($\oint P(\theta) \cos\theta dS = W$, where $dS$ represents an infinitesimal surface element), and the sum of the horizontal forces is zero ($\oint P(\theta)\sin\theta dS = 0$).

The surface pressure on the buttocks when seated on a flat cushion follows a characteristic distribution [30]—there is a pressure peak beneath each ischial tuberosity which gradually reduces to zero towards the periphery of the contact area (S1 Appendix). This contact pressure was modelled as a Gaussian distribution (see S1 Appendix). The spread of the pressure peak ($\alpha$) was varied between 0.2 and 0.35, which represent cushions with stiffness values beyond the range of those tested in section 2.2. We then modelled an externally applied lateral pressure by defining a second Gaussian term. This was controlled by its spread ($\beta$), its magnitude ($P_L$), and its location ($\theta_0$). $\beta$ and $\theta_0$ were fixed (0.4 and $\pi/4$ respectively) and $P_L/P_V$ (the ratio of lateral pressure relative to under-body pressure) was varied from 0% to 75%. This led to 16 parameter combinations, described in Table 2. Each pressure field was then applied as a boundary condition to the soft-tissue finite element model, and peak stresses and strains were calculated.

2.4.1 Optimising the surface pressure distribution. We considered the pressure distribution of a body suspended in a fluid as an ideal support scenario [22], as it results in minimal deviatoric stress relative to dilatational stress (see **S1 Appendix**). We then optimised the location ($\theta_0$) and relative magnitude of the lateral pressure ($P_L/P_V$) to minimise the difference between $P(\theta)$ and the distribution when suspended in a fluid, while ensuring that the full body weight (200N) was supported (see **S1 Appendix**).

## 3. Results

### 3.1 Applying lateral pressure reduces soft tissue deformations

We first set out to determine the effect of adding lateral pressure to a person seated on a standard support surface. We used a finite element model of the pelvis to simulate weight bearing while sitting on a soft cushion. Firstly, we simulated weight bearing without lateral support. In agreement with other studies [14,22,31], the model predicts significant stress concentrations

**Table 2. Parameters governing the spread of the under-body pressure ($\alpha$) and the magnitude of lateral pressure.**
Lateral pressure was defined relative to the peak under-body pressure ($P_L/P_V$).

| Pressure re-distribution ($\alpha$) | Lateral Pressure ($P_L/P_V$) |
|---|---|
| 0.2 | 0 |
| 0.25 | 0 |
| 0.3 | 0 |
| 0.35 | 0 |
| 0.2 | 0.25 |
| 0.25 | 0.25 |
| 0.3 | 0.25 |
| 0.35 | 0.25 |
| 0.2 | 0.50 |
| 0.25 | 0.50 |
| 0.3 | 0.50 |
| 0.35 | 0.50 |
| 0.2 | 0.75 |
| 0.25 | 0.75 |
| 0.3 | 0.75 |
| 0.35 | 0.75 |

under the ischial tuberosity (Fig 3A), with peak von Mises stresses of 58 kPa produced in the muscle. Then, when lateral pressure is applied, the peak von Mises stress beneath the bony prominence drops to 18 kPa, in support of our hypothesis. Contour plots show the stress is more evenly distributed in the soft tissues (Fig 3A), suggesting that more of the soft tissue is being recruited in transferring the load.

The volume of muscle tissue around the ischial tuberosity exposed to high von Mises stress ($> 20$ kPa) is reduced from 58% to 4% with lateral pressure application (Fig 3B). The volume of muscle tissue exposed to high shear strains ($> 0.2$) fell from 20% to 0%. Adding lateral pressure reduces the mean von Mises stress by 64% (95% CI 61% to 68%), while mean shear strains are reduced by 42% (95% CI 41% to 43%) when lateral pressure is applied (Fig 3B).

Plots of stress along a path through the soft tissue show that von Mises stress is reduced in all tissues under the ischial tuberosity when lateral pressure is applied (Fig 3C), with stress reduction being most pronounced in the muscle (66% in muscle, 43% in fat and 49% in skin). These results show that adding lateral pressure can reduce deep tissue deviatoric stress and deformation.

## 3.2 Applying lateral pressure reduces deformations to a greater extent than changing cushion stiffness

Having established that applying lateral pressure reduces deep tissue stress and deformation, we next assessed the effect of this intervention compared to a common device design consideration—changing cushion stiffness (Fig 4). Cushion stiffness is usually manipulated to reduce peak pressures by maximising the contact area with the soft tissue. Indeed, the soft cushion provides approximately 3.5 times more contact area than the stiff cushion (72 cm$^2$ in the stiff cushion, 177 cm$^2$ in the medium cushion and 255 cm$^2$ in the soft cushion), indicating that we are capturing a broad range of support surface stiffnesses. This area increase could be expected to achieve a similar reduction in deep tissue deformation. However, von Mises stresses (Fig 4A) and shear strains (Fig 4B) in the deep tissue are relatively less affected—with the change from a stiff to a soft cushion we see a reduction by a factor of 1.4 in peak von Mises stress (Fig

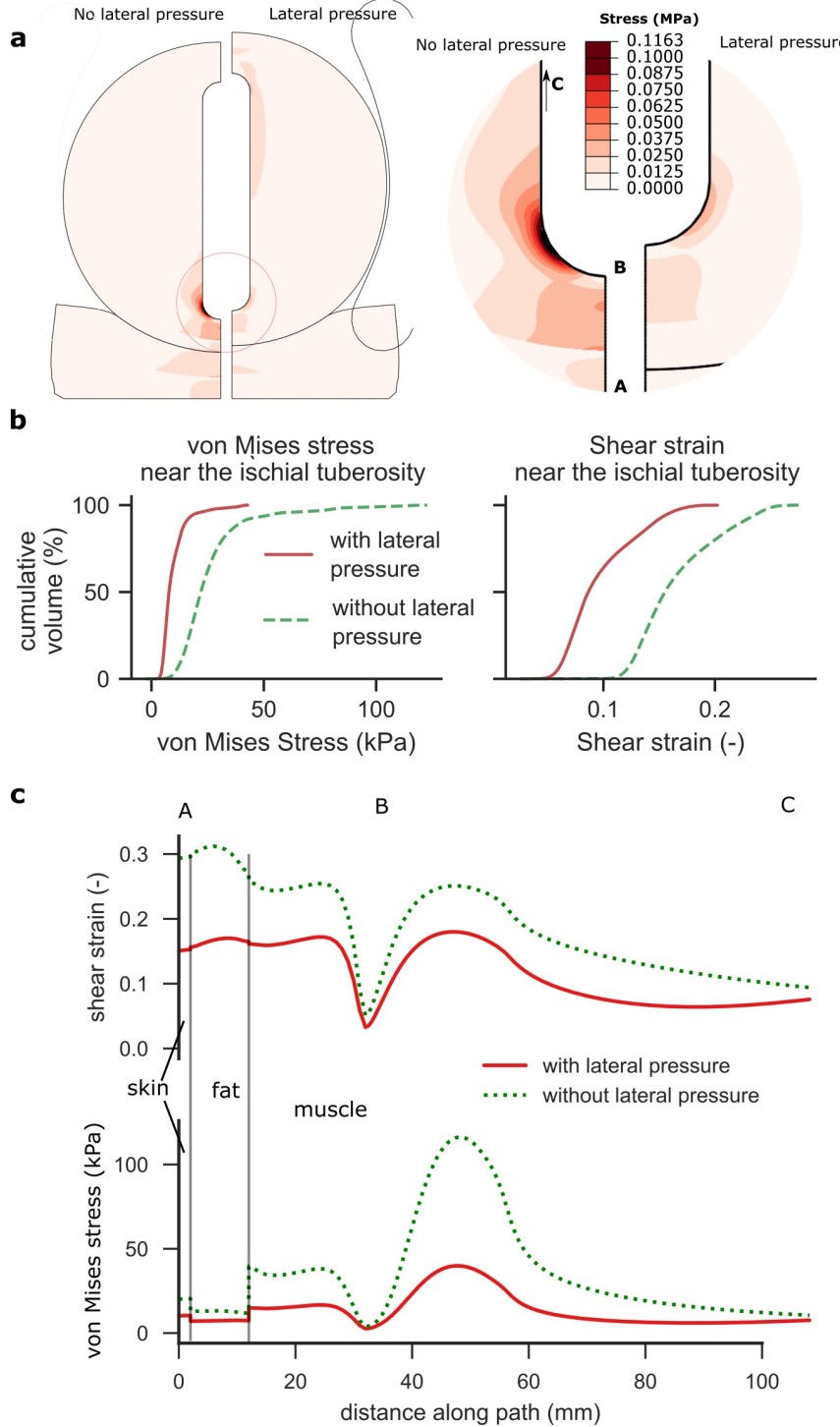

**Fig 3. Analysis of load-bearing when seated on a soft cushion.** In the absence of lateral pressure, the model predicts high von Mises stresses under the ischial tuberosity (a). With the introduction of lateral pressure (44 kPa chamber pressure), the region of high stress shrinks dramatically. Histograms of stresses and strains in the muscle tissue within a radius of 30 mm from the ischial tuberosity (b) indicate that von Mises stresses and shear strains are reduced. Analysis of the stress along path ABC (c) show a drop in von Mises stress and shear strain at the bony prominence, and throughout the muscle tissue. Shear strain and von Mises stress are also reduced in the skin and fat layers.

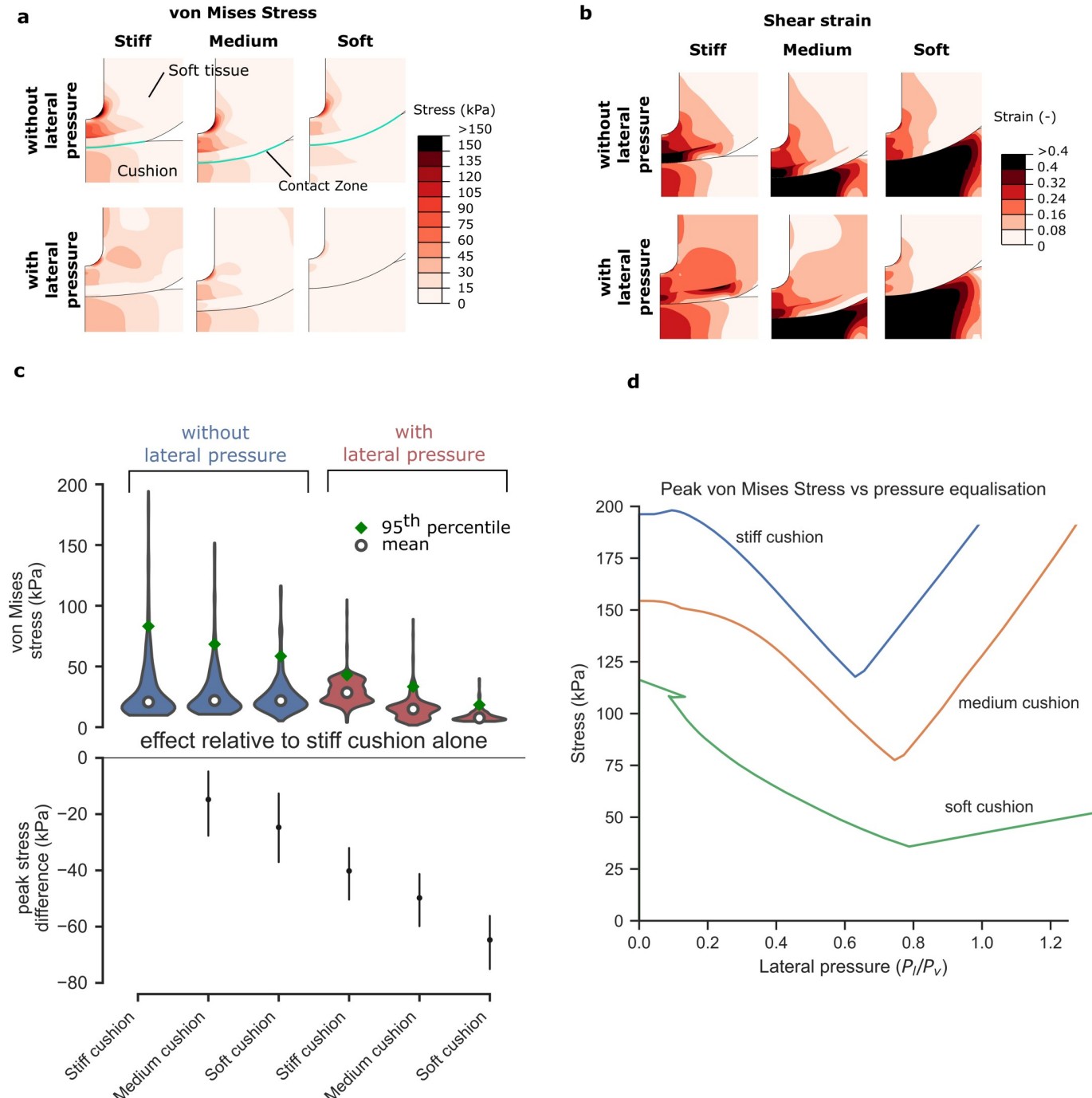

**Fig 4. Applying lateral pressure is more effective than changing cushion stiffness.** While the contact area varies substantially with cushion stiffness, the pattern of internal stress remains similar (a)—stress is concentrated at the bony prominence. Shear strains in the fat and skin are lower when a softer cushion is used (b), but strains within the muscle remain high for all cushions. These strains are reduced when lateral pressure is introduced. All three cushions benefit from the introduction of lateral pressure, with a soft cushion and lateral pressure providing the lowest von Mises stresses (c) [Violin plots show mean and 95th percentile values, stress difference plot shows the peak difference relative to a stiff cushion only with 95% confidence intervals]. As lateral pressure is gradually increased, the von Mises stress decreases until an optimum pressure is reached (d); beyond this pressure, von Mises stresses begin to increase again. While the magnitude of the optimum lateral pressure is different for each cushion, the ratio of lateral to vertical pressure is between 0.63 and 0.79 for all cushions tested.

4C). Without lateral pressure, all cushions induce a peak von Mises stress > 50 kPa. We find that introducing lateral pressure reduces the peak von Mises stresses observed with each cushion by a factor of 2.4 on average (1.9 for the stiff cushion, 2.1 for the medium cushion and 3.2 for the soft cushion; Fig 4C). The lowest peak deep tissue stresses are observed when the soft cushion is combined with lateral pressure, which reduces the peak von Mises stress to 18 kPa, suggesting a synergistic effect of combining lateral pressure with a soft cushion (Fig 4C).

We noticed that there is an optimum magnitude of lateral pressure which is different for each cushion (38.5 kPa, 37.9 kPa and 12.2 kPa for stiff, medium and soft cushions, respectively); however, the ratio of lateral to under-body pressure is consistently between 0.6 and 0.8 (Fig 4D). This suggests that balancing under-body and lateral pressures is more important for the reduction of deep tissue deviatoric stress than reducing peak under-body pressures.

## 3.3 Lateral pressure reduces stresses in a 3D model

Upon addition of lateral pressure, stresses were reduced at the ischial tuberosities in a similar way to the 2D model (Fig 5A). The volume of soft tissue exposed to high von Mises stress greatly reduced when lateral pressure was applied (Fig 5B). Adding lateral pressure reduced peak von Mises stress in the muscle by a factor of 2.5 when a stiff cushion was used, 2.6 with a medium-stiffness cushion and 2.4 with a soft cushion (Fig 5C). With optimal lateral pressure applied, the stresses at the greater trochanter and bones of the hemipelvis reached no more than 22% of the load at the ischial tuberosity. These results demonstrate that the effects found in 2D are representative of the 3D environment. They also show that gentle lateral pressure can be applied without compromising the tissue at the femur or sacrum. It should be noted that the lateral device modelled here was a very simple design, and no optimisation of its shape was performed. By contouring the lateral pressure device or other optimisations, further efficacy may be possible.

## 3.4 Surface pressure equalisation is necessary to protect deep tissue from deformation

Having found that deep tissue deformations were minimised for each cushion when under-body and lateral pressure were in a specific ratio (0.6–0.8), we sought to test whether this ratio could form the basis of a design principle. We removed the cushion from the model and replaced it with a surface pressure boundary condition that could be manipulated independently of cushion design. We varied both the ratio of lateral to under-body pressure ($P_L/P_V$), and the spread of underbody pressure ($\alpha$) and measured peak von Mises stress for all combinations (Table 2).

As in the previous analyses, in the absence of lateral pressure, redistributing under-body pressure (increasing the spread, $\alpha$, of the pressure peak), reduces peak von Mises stresses at the ischial tuberosity, but even substantial re-distribution fails to reduce the stress below 100 kPa (112 kPa is observed when $\alpha = 0.35$; Fig 6A). In contrast, inducing a lateral to under-body pressure ratio of 0.25 reduces peak von Mises stress from 180 kPa to 67 kPa. The presence of lateral pressure appears to reduce the effect of redistributing under-body pressure (Fig 6A), suggesting that when lateral pressure is employed, it becomes the most important factor in reducing deformations. These results indicate that controlling the ratio of lateral to under-body pressure ($P_L/P_V$) is necessary to achieve low deep-tissue stress.

To understand why lateral pressure may be critical, we studied how this ratio affects the shape of the pressure distribution when compared to two extreme scenarios: the pressure distribution while sitting on a stiff cushion (a high-deformation scenario), and that when suspended in a fluid (a low-deformation scenario). The shape of these distributions is markedly

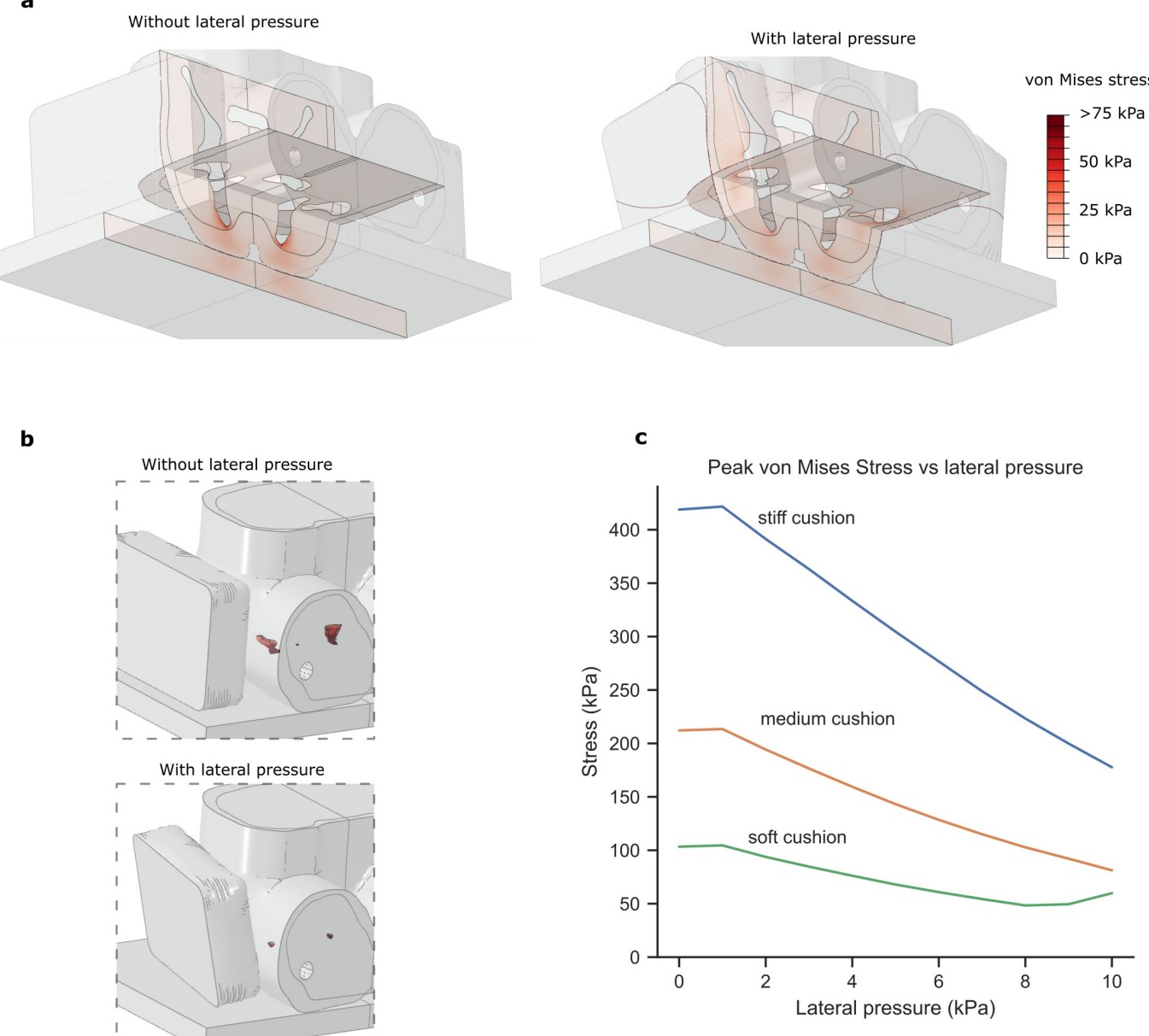

**Fig 5. A 3D model of the seated pelvis under load.** (a) Results for the stiff cushion shown with and without lateral pressure applied. Coronal and transverse sections are shown to indicate von Mises stresses both at the ischial tuberosities and the greater trochanter. (b) The volume of soft tissue exposed to high stresses (>32kPa) is shown in relation to the whole pelvis. The whole pelvis is made transparent to help visualise the location of high stresses (beneath the ischial tuberosity) (c) Change in peak von Mises stress throughout the soft tissue of the pelvis (surrounding both the ischium and the femur) as lateral pressure is increased.

different (Fig 1A in S1 Appendix), with a sharp peak of pressure beneath the ischial tuberosity when seated on a cushion, versus a smooth, even pressure distribution when submersed. A parametric study showed that adding lateral pressure best mimicked the pressure profile of suspension in a fluid (S1 Appendix).

We then optimised the magnitude and angle of the lateral pressure to best mimic suspension in a fluid. Contour plots show that optimising this lateral pressure (to $P_L/P_V = 0.71$ and $\theta_0 = 61.5°$) can mimic the internal stresses experienced while suspended in a fluid (Fig 6B). In

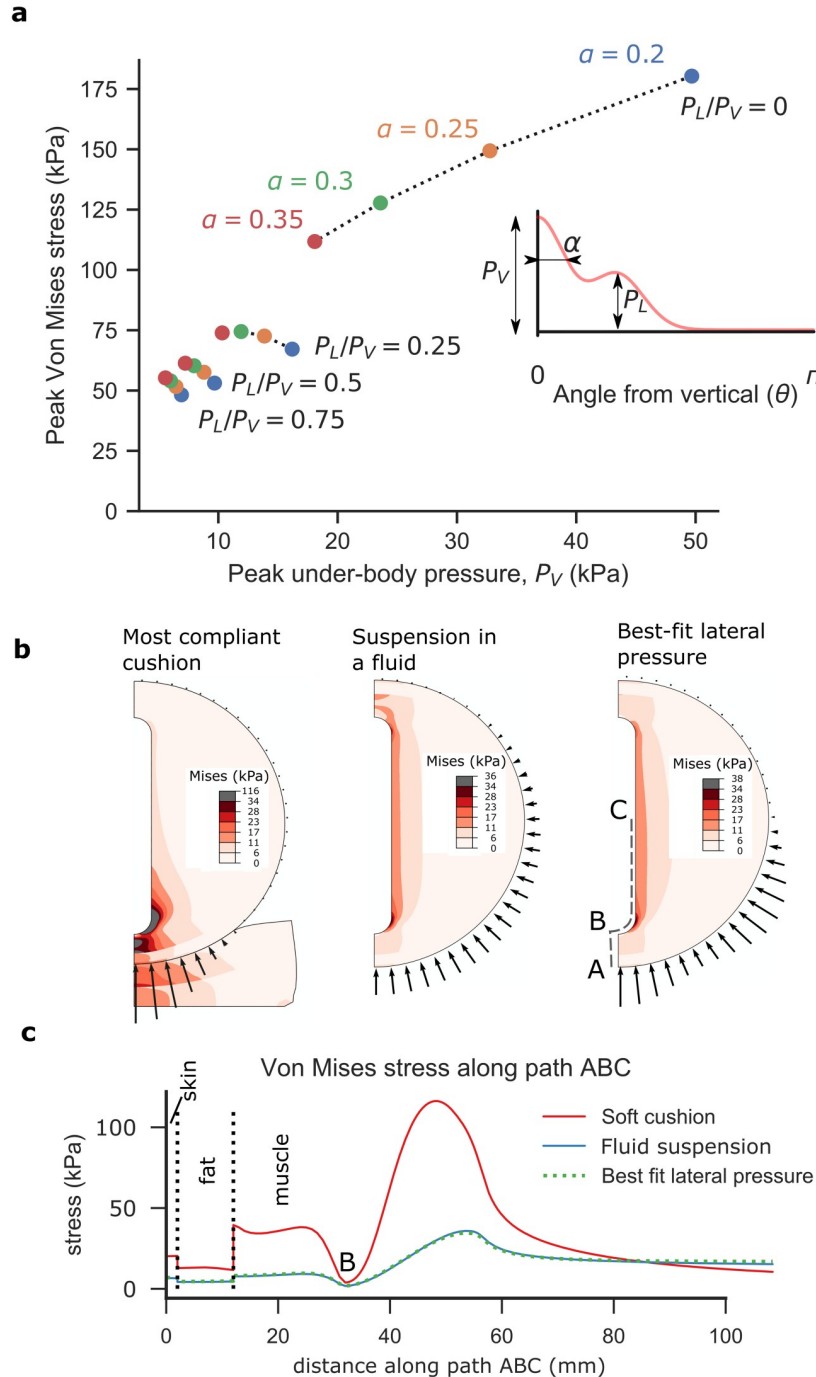

**Fig 6. Surface pressure analysis.** Redistributing under-body pressure ($P_V$) reduces peak von Mises stresses when no lateral pressure is applied (a), but peak stresses remain above 100 kPa. Counter-acting that pressure with a lateral pressure ($P_L$) reduces peak stresses to a greater extent. When the magnitude and angle of lateral pressure is optimised, the deep tissue von Mises stresses approach that of suspension in a fluid (b; arrows illustrate pressure intensity). Path plots of von Mises stress show that lateral pressure can induce a similar stress profile at the bony prominence to that when suspended in a fluid (c).

addition, with these parameters, von Mises stresses at the ischial tuberosity were either equal to or less than those induced when suspended in a fluid (Fig 6C).

In summary, not only can applying lateral pressure reduce deep tissue von Mises stress and deformation, we have found that an optimal magnitude and location of lateral pressure can mimic the environment induced when suspended in a fluid.

## 4. Discussion

The goal of reducing peak surface pressures at vulnerable body sites has underpinned the design of almost all medical support surfaces to date. Meanwhile, studies have consistently concluded that peak surface pressures do not accurately predict internal tissue mechanics [11,14], nor are they effective in predicting patients at risk of pressure ulcers [10]. In this study, we have shown that ensuring under-body and lateral pressures are balanced—a principle we call pressure equalisation—is more effective at reducing deep tissue deformations than reducing peak under-body pressure. We postulate that devices designed to maintain a prescribed ratio of lateral pressure to under-body pressure will reduce the risk of pressure ulcer formation in the soft tissue of immobile patients.

The shift in emphasis from pressure re-distribution to pressure equalisation has implications for support surface design (Fig 7). The synthesised results of multiple clinical trials [8,9] suggest that any well-designed mattress is better than a standard hospital bed, but none are particularly successful at reducing pressure ulcer risk. Pressure redistributing devices (either passive or active, Fig 7A and 7B respectively) may protect against superficial ulcers, while having little effect on deep tissue injuries [14]. Our results show that devices must be capable of providing sufficient lateral support to counter-act the deformations induced by under-body pressure. Immersion/encapsulation-based devices such as water beds [32] aim to increase the contact area between the soft tissue and the support surface; however, while the contact area may increase, the horizontal pressures at the periphery of the contact area are usually minimal (Fig 7C), as pressure is primarily in reaction to gravitational body force. As this force acts in the vertical direction, it would be insufficient to equalise the under-body pressures and prevent bulging. Devices that aim to directly minimise shape change have also been developed [33], but their dependence on vertical translations of pistons means that they are not suited to regulating lateral pressure. We believe that support surface technology may yet reduce pressure ulcer prevalence if they are redirected to achieving pressure equalisation.

The pressure equalisation principle has implications for pressure measurement as a diagnostic tool and as a method of evaluating support surfaces. Surfaces incorporating arrays of pressure sensors have been suggested as early-warning systems for ulceration [34,35], and are frequently used to evaluate new support surfaces [36–38]. Using this technology, devices can be readily differentiated based on the peak pressures they produce. However, when these devices are then compared through clinical outcomes, the differences between them vanish [8,9], and so the current predictive power of pressure measurement is limited. If surface pressure could be measured all around the soft tissues (Fig 7D), then the level of pressure equalisation may be a more predictive tool. Then, a measure of the ratio of lateral to under-body pressure ($P_L/P_V$) could be used to determine ulcer risk, and as a control signal for active devices.

The pressure gradient, defined as the spatial change in pressure from the point of peak pressure, has been proposed as an alternative to peak pressure for predicting soft tissue damage [39]. At first glance, pressure equalisation may seem to be equivalent to using pressure gradient as an ulceration indicator. However, the pressure gradient does not account for the direction of pressure, and so a body could be loaded with a low pressure gradient yet have little or no pressure equalisation, because lateral pressures are not considered. In other words, pressure gradient is a local variable, as is peak pressure, whereas pressure equalisation is a measure of the quality of the body support as a whole.

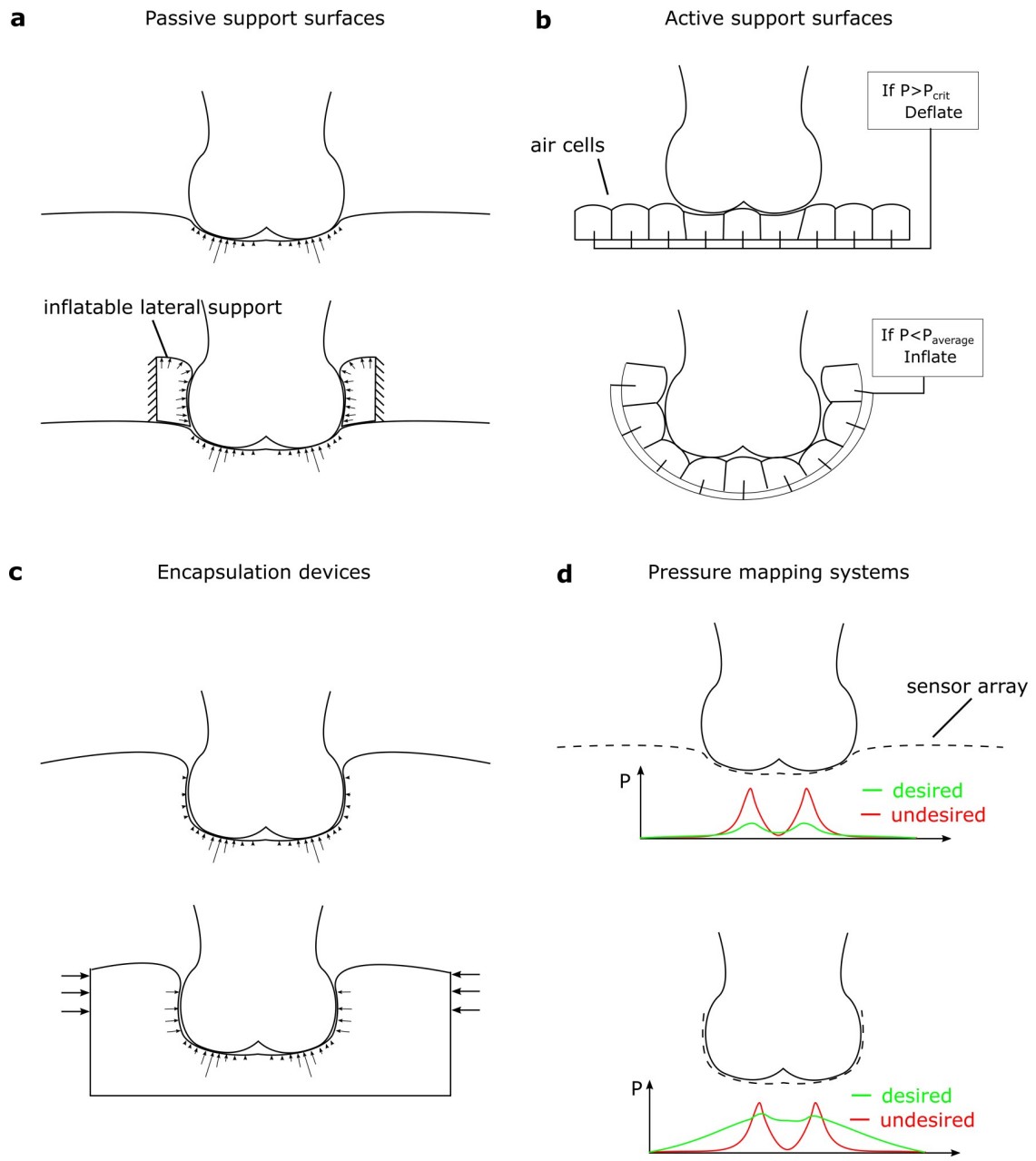

**Fig 7. Pressure equalisation and its effects on device design.** For surfaces designed to reduce peak pressure passively (a), applying a lateral pressure device helps to avoid lateral bulging (top, showing current devices, bottom showing improved design). Active devices based on individually controlled air cells (b) could be improved by surrounding the soft tissue and changing the control software to aim for equalised pressure, rather than reduce peak pressure. Encapsulation devices achieve large contact areas, but the lateral pressures exerted may be limited (c). These could be improved by active compression or smart materials. Pressure mapping systems (d) currently identify pressure peaks as undesirable. If they could measure pressure around the surface, then they could be re-purposed to measure the level of pressure equalisation.

From a clinical perspective, devices that can equalise under-body pressure with lateral pressure may be a vital tool in reducing the incidences of pressure ulcers—both hospital-acquired ulcers and those that are acquired in community care settings. The ability to apply sufficient lateral pressure will, however, need to be balanced with other equally important design considerations. For example, wheelchair users must not be exposed to intermittent high lateral

pressures as they move in the chair. This inflexibility is one reason why form-fitted cushions are not a solution to preventing tissue distortion. In contrast, a successful device must be flexible enough to provide a well-distributed and well-controlled lateral pressure regardless of patient movement. Other design considerations include regulating the temperature and humidity at the skin surface, as well as ease of installation, use and cleaning. While this work has focused on the biomechanics of lateral pressure in general, the practical application of this principle will be more complex and require significant innovation in device design.

The reductions in deep tissue stress and strain possible through surface pressure equalisation could be sufficient to reduce pressure ulcer risk. The safe magnitude of deformation (and even the most appropriate measure) is not yet fully accounted for [40] and it is likely to be patient, environment, and tissue-specific. In this study, we have used two measures of deep tissue mechanics—von Mises stress and shear strain. These measures aim to capture the deformations likely to lead to capillary and lymphatic vessel restriction, and cell deformation, which contribute to pressure ulcer onset. Experiments using rat muscle under compression [41] indicated that stresses greater than 32 kPa induced damage, with this threshold dropping to 9 kPa over prolonged loading, while work quantifying deformations in a similar model [42] indicated that damage occurred above a shear strain of 0.3. Our results indicate that redistributing under-body pressure would not protect soft tissue from these levels of deformation, but that applying lateral pressure could.

The 2D finite element model used here simplified the anatomical structure of the pelvis in a similar way to previous studies [14,22,31]. These idealisations allowed us to focus on the general case of a bony prominence transferring load through soft tissue to a support surface, and enabled the comparisons and analyses described here. The 3D model used here were important to support the conclusions drawn from 2D analyses, but there are also limitations to this model. While we have included more anatomical complexity including thighs, femurs and pelvis, more biofidelic models have been proposed [43–45]. For example, we chose tissue mechanical properties in line with Oomens et al. [14], but there are several published models of soft tissue mechanical properties that vary in complexity [24–27]. This makes conclusions based on absolute stress values difficult. A further complexity not accounted for in our model is the posture and secondary supports (arm rests, for example) of the patient, which may affect the loading boundary conditions. For these reasons, we have focused on the relative effects of interventions on stresses and strains, thus making the conclusions robust against the chosen material models and boundary conditions. Using more biofidelic approaches will be a key step in applying the current results in the clinic. In particular, models generated from high-risk patients as opposed to healthy volunteers will be crucial. Physical validation will need to come from measurements of internal tissue deformations, for example through load-bearing MRI [46,47].

Our results suggest a novel method for creating a safer mechanical environment. We hypothesize that the reduced deformations created by lateral pressure equalization will translate into better deep tissue blood perfusion and lower risk of deformation-induced cell damage. A key next step will be to test this hypothesis by measuring the physiological response of the soft tissues of seated patients, for example through measuring transcutaneous gas tension [34].

In conclusion, a change in focus from redistributing under-body pressure to equalising it with lateral pressure will lead to new innovations and improvements to patient care, resulting in a reduction of pressure ulcer prevalence in immobile patients.

## Supporting information

**S1 Appendix. Analysis of pressure distributions.**
(PDF)

**S2 Appendix. Testing model assumptions.**
(PDF)

# Acknowledgments

We would like to thank Surbhi Gupta and all those involved at Imperial Innovations Ltd. for their support in delivering this project.

# Author Contributions

**Conceptualization:** Colin J. Boyle, Claire A. Higgins, Marc A. Masen, Spyros D. Masouros.

**Data curation:** Colin J. Boyle.

**Formal analysis:** Colin J. Boyle, Diagarajen Carpanen, Thanyani Pandelani.

**Funding acquisition:** Colin J. Boyle, Claire A. Higgins, Marc A. Masen, Spyros D. Masouros.

**Investigation:** Colin J. Boyle.

**Methodology:** Colin J. Boyle, Diagarajen Carpanen, Thanyani Pandelani.

**Project administration:** Spyros D. Masouros.

**Supervision:** Claire A. Higgins, Marc A. Masen, Spyros D. Masouros.

**Visualization:** Colin J. Boyle.

**Writing – original draft:** Colin J. Boyle.

**Writing – review & editing:** Diagarajen Carpanen, Thanyani Pandelani, Claire A. Higgins, Marc A. Masen, Spyros D. Masouros.

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
