## [Decision Letter · Decision Letter 0]

19 Nov 2019

PONE-D-19-27579

Lateral pressure equalisation as a principle for designing support surfaces to
prevent deep tissue pressure ulcers

PLOS ONE

Dear Dr Boyle,

Thank you for submitting your manuscript to PLOS ONE. After careful consideration, we
feel that it has merit but does not fully meet PLOS ONE’s publication criteria as it
currently stands. Therefore, we invite you to submit a revised version of the
manuscript that addresses the points raised during the review process.

We would appreciate receiving your revised manuscript by Jan 03 2020 11:59PM. When
you are ready to submit your revision, log on to https://www.editorialmanager.com/pone/ and select the 'Submissions
Needing Revision' folder to locate your manuscript file.

If you would like to make changes to your financial disclosure, please include your
updated statement in your cover letter.

To enhance the reproducibility of your results, we recommend that if applicable you
deposit your laboratory protocols in protocols.io, where a protocol can be assigned
its own identifier (DOI) such that it can be cited independently in the future. For
instructions see: http://journals.plos.org/plosone/s/submission-guidelines#loc-laboratory-protocols

We look forward to receiving your revised manuscript.

Kind regards,

Pedro H. Oliveira, Ph.D.

Academic Editor

PLOS ONE

2. Thank you for providing the following Funding Statement: 

"This project was partly funded by an Imperial College London and Imperial
Innovations Ltd. Proof of Concept funding to CB and SM and EPSRC funding to CH, MM
and SM (EP/N026845/1). The funders had no role in study design, data collection and
analysis, decision to publish, or preparation of the manuscript.".

We note that one or more of the authors is affiliated with the funding organization,
indicating the funder may have had some role in the design, data collection,
analysis or preparation of your manuscript for publication; in other words, the
funder played an indirect role through the participation of the co-authors.

If the funding organization did not play a role in the study design, data collection
and analysis, decision to publish, or preparation of the manuscript and only
provided financial support in the form of authors' salaries and/or research
materials, please review your statements relating to the author contributions, and
ensure you have specifically and accurately indicated the role(s) that these authors
had in your study in the Author Contributions section of the online submission form.
Please make any necessary amendments directly within this section of the online
submission form.  Please also update your Funding Statement to include the following
statement: “The funder provided support in the form of salaries for authors [insert
relevant initials], but did not have any additional role in the study design, data
collection and analysis, decision to publish, or preparation of the manuscript. The
specific roles of these authors are articulated in the ‘author contributions’
section.”

If the funding organization did have an additional role, please state and explain
that role within your Funding Statement.

Please also provide an updated Competing Interests Statement declaring this
commercial affiliation along with any other relevant declarations relating to
employment, consultancy, patents, products in development, or marketed products,
etc.  

3. We note that you have a patent relating to material pertinent to this article.
Please provide an amended statement of Competing Interests to declare this patent
(with details including name and number), along with any other relevant declarations
relating to employment, consultancy, patents, products in development or modified
products etc. Please confirm that this does not alter your adherence to all PLOS ONE
policies on sharing data and materials, as detailed online in our guide for authors
http://journals.plos.org/plosone/s/competing-interests by including
the following statement: "This does not alter our adherence to  PLOS ONE policies on
sharing data and materials.” If there are restrictions on sharing of data and/or
materials, please state these. Please note that we cannot proceed with consideration
of your article until this information has been declared.

Reviewers' comments:

Reviewer's Responses to Questions

**Comments to the Author**

1. Is the manuscript technically sound, and do the data support the conclusions?

Reviewer #1: Yes

Reviewer #2: Yes

2. Has the statistical analysis been performed
appropriately and rigorously? 

Reviewer #1: N/A

Reviewer #2: Yes

3. Have the authors made all data underlying the
findings in their manuscript fully available?

Reviewer #1: Yes

Reviewer #2: Yes

4. Is the manuscript presented in an intelligible
fashion and written in standard English?

Reviewer #1: Yes

Reviewer #2: Yes

5. Review Comments to the Author

Reviewer #1: The paper deals with a basic research regarding the opportunity or not
to use lateral pressure to be applied to the patient forced to maintain a sitting
position for long periods in order to reduce the pressure at the interface with the
pillow and consequently the risk of ulcerations.

The idea is good and its applicability must be verified. The preliminary study is
well written and the methods used are appropriate.

Small improvements are required:

- always define all the parameters introduced (for example in table 1 and in the
equations)

- in paragraph 2.1.2 it is well known that the body mass of the torso + head + upper
limbs is equal to 2/3 of the whole body mass not 1/2

- in paragraph 2.1.2 next to the figure 80 the unit of measurement is missing

- in brackets, after the softwares used, it is necessary to mention the company that
distributes them and their location

- in paragraph 2.3 justify the choices made on the imposed loads.

Reviewer #2: This study sought to present a novel approach to prevent ulcers due to
deep tissue pressure. In particular the authors stated that adding a supporting
lateral pressure (they called “pressure equalization”) to the classical under-body
pressure redistribution approach could provide a more effective solution to reduce
deep tissue deformation that can lead to ulcers development. The work is based on
Finite Element (FE) Model, which is able to predict the stresses induced on the deep
tissue; the authors used this model to assess the differences introduced by the
lateral pressure equalization. In particular, they found that managing the ratio of
peak lateral pressure to peak under-body pressure and the position of the lateral
supports could be used to regulate deep tissue stresses. The authors stated that the
overall deformation can be decreased at the induced when the body is suspended in a
fluid. The approach can be used to design and assess novel solutions related to
supporting devices.

General Comments

The main working hypothesis at the basis of this paper is clearly reported as far as
the main objective. The introduction to the approach is clearly reported and the
modelling phase well documented. Although the methodology is not that innovative,
the application to is indeed original. I guess the audience of PLOS ONE will
appreciate this work.

The structure of the article seems to be precise (Abstract, Introduction, Methods
[with subheadings], Results, Discussion).

Modelling and data analysis seem to be clearly reported and coherent with the work
objectives. Several minor concerns have been already reported to the authors.

The use of the English language seems to be correct.

The references to previous works seem to be precise and up-to-date.

Specific Comments

Abstract

In general this section is ok. Please, could you report also the magnitude of the
deformation related to deep tissue and the differences between under-body pressure
redistribution and lateral pressure equalization?

Introduction

Really well written. You well reported the main clinical problem, the
state-of-the-art related to the main provided solutions, your hypothesis and the
rationale behind your work. Really appreciated Figure 1.

Methods

• Page 8. Model properties. Oomens’ work is a modelling approach dating back to 2003.
Is there any more recent work identifying soft tissues properties and modelling?

• Page 11 – 2.1.2 Line 6. I guess that the unit of measurements is missing.

• Please provide more hypothesis on the choice of not introducing any friction
between the support surface and the skin. Is this condition the real one?

• Page 12. Please provide more information about the “adjusting” of data accounting
for volume variation. It is not that clear.

• Page 16. Table 2. Please provide information about “[0.5ex]”, present in the first
line.

Results

Very well reported both graphically and in the main text.

Discussion

In general, this section is ok. Since you reported also the most clinical
perspective, what about pressure equalization on vessels? Could you speculate on any
main issue?

6. PLOS authors have the option to publish the peer
review history of their article (what does this mean?). If published, this will
include your full peer review and any attached files.

If you choose “no”, your identity will remain anonymous but your review may still be
made public.

**Do you want your identity to be public for this peer review?** For
information about this choice, including consent withdrawal, please see our
Privacy Policy.

Reviewer #1: No

Reviewer #2: Yes: Nicola Francesco Lopomo

---

## [Author Response · Author response to Decision Letter 0]

21 Nov 2019

We wish to thank the reviewers for their valuable critique of our manuscript. Pease
see below for our responses to each comment. 

Reviewer #1: The paper deals with a basic research regarding the opportunity or not
to use lateral pressure to be applied to the patient forced to maintain a sitting
position for long periods in order to reduce the pressure at the interface with the
pillow and consequently the risk of ulcerations.

The idea is good and its applicability must be verified. The preliminary study is
well written and the methods used are appropriate.

Small improvements are required:

- always define all the parameters introduced (for example in table 1 and in the
equations)

We have reviewed the manuscript to ensure all parameters are now defined at the point
of first instance.

- in paragraph 2.1.2 it is well known that the body mass of the torso + head + upper
limbs is equal to 2/3 of the whole body mass not ½

We agree that this 50% value does not represent all upper-body weight. Oomens et al
[14] discuss loading boundary conditions in pelvis models. Using literature on
experimental pressure-reading, they describe that weight borne by the IT during
sitting varies from 18% to 77% of body weight. This variation is due to patient
posture, including use of arm and thigh support, and inclination of back support. We
chose 50% as a value within the experimental range, while also enabling direct
comparison with that previous modelling work [14]. 

We also argue that because we look at relative differences (between support
surfaces), rather than draw conclusions based on absolute values, the choice of load
magnitude does not affect our conclusions.

Changes to text:

Section 2.1.2 pg 8:

The proportion of body weight borne by the ischial tuberosities while seated varies
from 18% to 77% [14]. We estimated the amount of load supported by the pelvis at
400N — representing approximately 50% of the body weight of an 80 kg adult (with
each tuberosity bearing 200N) because it is within the range of experimental
findings and enables direct comparison with Oomens et al [14].

Discussion pg 25:

A further complexity not accounted for in our model is the posture and secondary
supports (arm rests, for example) of the patient, which may affect the loading
boundary conditions. For these reasons, we have focused on the relative effects of
interventions on stresses and strains, thus making the conclusions robust against
the chosen material models and boundary conditions. Using more biofidelic approaches
will be a key step in applying the current results in the clinic.

- in paragraph 2.1.2 next to the figure 80 the unit of measurement is missing

Thank you, amended with kPa as units

- in brackets, after the softwares used, it is necessary to mention the company that
distributes them and their location

Amended as suggested

- in paragraph 2.3 justify the choices made on the imposed loads.

Our objective for developing this 3D model was to ensure that the conclusions drawn
from the axisymmetric model translated to a 3D environment. To achieve this
objective, we modelled the same loading regime as the axisymmetric model. We accept,
however, that a more biofidelic model – based on realistic load cases including body
freedom-of-movement and whole-body simulation – will be required as a next step
before translation to the clinic. 

Changes to text (the alterations to section 2.1.2 are also relevant here):

Section 2.3 pg 11:

A body force of 200N was applied to the bone nodes. This represents a full body
weight of 80 kg, with the assumption that 50% of this travels through the pelvis of
a seated individual (as was assumed with the axisymmetric model and is based on
Oomens et al. [14], see section 2.1.2). Maintaining this level of loading in the 3D
model aids comparison between that and the axisymmetric model.

Reviewer #2: This study sought to present a novel approach to prevent ulcers due to
deep tissue pressure. In particular the authors stated that adding a supporting
lateral pressure (they called “pressure equalization”) to the classical under-body
pressure redistribution approach could provide a more effective solution to reduce
deep tissue deformation that can lead to ulcers development. The work is based on
Finite Element (FE) Model, which is able to predict the stresses induced on the deep
tissue; the authors used this model to assess the differences introduced by the
lateral pressure equalization. In particular, they found that managing the ratio of
peak lateral pressure to peak under-body pressure and the position of the lateral
supports could be used to regulate deep tissue stresses. The authors stated that the
overall deformation can be decreased at the induced when the body is suspended in a
fluid. The approach can be used to design and assess novel solutions related to
supporting devices.

General Comments

The main working hypothesis at the basis of this paper is clearly reported as far as
the main objective. The introduction to the approach is clearly reported and the
modelling phase well documented. Although the methodology is not that innovative,
the application to is indeed original. I guess the audience of PLOS ONE will
appreciate this work.

The structure of the article seems to be precise (Abstract, Introduction, Methods
[with subheadings], Results, Discussion).

Modelling and data analysis seem to be clearly reported and coherent with the work
objectives. Several minor concerns have been already reported to the authors.

The use of the English language seems to be correct.

The references to previous works seem to be precise and up-to-date.

Specific Comments

Abstract

In general this section is ok. Please, could you report also the magnitude of the
deformation related to deep tissue and the differences between under-body pressure
redistribution and lateral pressure equalization?

Thank you for this suggestion, we have updated the abstract to include magnitudes as
well as fold-change.

Changes to text:

Abstract: 

A finite element model of the seated pelvis predicts that applying a lateral pressure
to the soft tissue reduces peak von Mises stress in the deep tissue by a factor of
2.4relative to a standard cushion (from 113 kPa to 47 kPA) — a greater effect than
that achieved by using a more conformable cushion, which reduced von Mises stress to
75 kPa. Combining both a conformable cushion and lateral pressure reduced peak von
Mises stresses to 25 kPa. 

Introduction

Really well written. You well reported the main clinical problem, the
state-of-the-art related to the main provided solutions, your hypothesis and the
rationale behind your work. Really appreciated Figure 1.

Methods

• Page 8. Model properties. Oomens’ work is a modelling approach dating back to 2003.
Is there any more recent work identifying soft tissues properties and modelling?

Thank you for this comment. We accept that the material properties chose may not be
the definitive material models for each of the soft tissues. For example, more
complex models that incorporate transverse isotropy of muscle/skin tissue or
viscoelasticity may yield more accurate predictions of absolute stress/strain
magnitudes. However, our study focuses on relative differences between groups (on
different cushions, and with/without lateral pressure). 

We argue that this focus on relative differences reduces the sensitivity of our
conclusions to specific material models, particularly because our chosen models
incorporate the two most important aspects of soft tissue mechanics
(large-deformations and material non-linearity). We also note that using these
models enabled direct comparison to the study most closely related to our work
(Oomens et al [14]). 

We also understand that incorporating more accurate material models of soft tissues
will be a crucial part of creating more biofidelic models, and that this is a next
step on the way to validating our pressure-equalisation approach for the clinic.

Changes to text:

Methods, page 7:

There have been many material models of skeletal muscle [24,25], skin [26], and to a
lesser extent fat [27]. However, experimentally-based models that quantify all three
tissues together are rarer, making it difficult to combine tissues defined from
different experimental setups. In this study, we used the material models based on
Oomens et al [14] because it enabled direct comparison with that study, and because
all three soft tissues were characterized. Each region was assigned an Ogden
hyperelastic material model, and parameter values are listed in Table 1.

Discussion, page 25:

For example, we chose tissue mechanical properties in line with Oomens et al. [14],
but there are several published models of soft tissue mechanical properties that
vary in complexity [24–27]. This makes conclusions based on absolute stress values
difficult. A further complexity not accounted for in our model is the posture and
secondary supports (arm rests, for example) of the patient, which may affect the
loading boundary conditions. For these reasons, we have focused on the relative
effects of interventions on stresses and strains, thus making the conclusions robust
against the chosen material models and boundary conditions. Using more biofidelic
approaches will be a key step in applying the current results in the clinic.

• Page 11 – 2.1.2 Line 6. I guess that the unit of measurements is missing.

Amended to include kPa as unit.

• Please provide more hypothesis on the choice of not introducing any friction
between the support surface and the skin. Is this condition the real one?

We used frictionless contact as a simplification of the real-world situation. To
assess the impact of this simplification, we ran a sensitivity study where the
effect of including friction on peak stresses and strains was quantified (see S2
Appendix). We found that in our load case, friction had negligible effect on peak
stresses. However, we accept that in future studies that aim to simulate more
biofidelic load cases, modelling friction effects will become increasingly
important. In particular, a model that captured patient posture dynamically would
require more accurate contact characterisation. 

We also note that our primary focus in this study was the deep tissue (the source of
the most dangerous pressure ulcers), while friction tends to induce superficial
pressure ulcers.

Changes to text:

Methods page 9:

While this is a simplification of the real-world scenario, a sensitivity study
revealed friction to have a negligible effect on model predicitons (S2
Appendix).

• Page 12. Please provide more information about the “adjusting” of data accounting
for volume variation. It is not that clear.

Thank you for this comment. We developed an analysis approach to ensure that our
conclusions were insensitive to the mesh density. Our aim was to summarise the
stress/strain in a tissue. One approach is to simply probe one point near the region
of interest. However, this is sensitive to the choice of point. Another approach is
to summarise all the points within a region of interest (mean, median, 95
percentile, etc). However, this approach is affected by mesh density, so locations
with many small elements contribute more to the summary statistic than locations
with fewer large elements. To overcome this limitation, we took a weighted sample of
all elements in the region of interest. The sample was weighted by the volume of the
element, so an element with double the volume is twice as likely to be sampled. 

Changes to text:

Page 10:

To summarise the stresses and strains in the deep tissue, we sampled 1600 elements
within a 30mm radius of the ischial tuberosity. Since elements vary considerably in
size throughout the region, we weighed our sampling by element volume (IVOL output
from ABAQUS). For example, an element with a volume 2V is twice as likely to be
sampled as an element with volume V. Weighting the results like this ensures that
analyses are independent of mesh density, which varies throughout the model.

• Page 16. Table 2. Please provide information about “[0.5ex]”, present in the first
line.

Thank you for spotting this, it is an artefact of a conversion from Latex to MS Word
and has been deleted.

Results

Very well reported both graphically and in the main text.

Discussion

In general, this section is ok. Since you reported also the most clinical
perspective, what about pressure equalization on vessels? Could you speculate on any
main issue?

If we understand your question correctly, you would like us to discuss how our
mechanical results translate into improvements to blood perfusion, thereby keeping
tissue oxygenated and uninjured. This is a great point, and we have added a
paragraph to the discussion.

Changes to text:

Discussion, Page 27:

Our results suggest a novel method for creating a safer mechanical environment. We
hypothesize that the reduced deformations created by lateral pressure equalization
will translate into better deep tissue blood perfusion and lower risk of
deformation-induced cell damage. A key next step will be to test this hypothesis by
measuring the physiological response of the soft tissues of seated patients, for
example through measuring transcutaneous gas tension [34].

---

## [Decision Letter · Decision Letter 1]

12 Dec 2019

Lateral pressure equalisation as a principle for designing support surfaces to
prevent deep tissue pressure ulcers

PONE-D-19-27579R1

Dear Dr. Boyle,

We are pleased to inform you that your manuscript has been judged scientifically
suitable for publication and will be formally accepted for publication once it
complies with all outstanding technical requirements.

With kind regards,

Pedro H. Oliveira, Ph.D.

Academic Editor

PLOS ONE

**Comments to the Author**

1. If the authors have adequately addressed your comments raised in a previous round
of review and you feel that this manuscript is now acceptable for publication, you
may indicate that here to bypass the “Comments to the Author” section, enter your
conflict of interest statement in the “Confidential to Editor” section, and submit
your "Accept" recommendation.

Reviewer #1: (No Response)

Reviewer #2: All comments have been addressed

2. Is the manuscript technically sound, and do the data
support the conclusions?

Reviewer #1: (No Response)

Reviewer #2: Yes

3. Has the statistical analysis been performed
appropriately and rigorously? 

Reviewer #1: (No Response)

Reviewer #2: Yes

4. Have the authors made all data underlying the
findings in their manuscript fully available?

Reviewer #1: (No Response)

Reviewer #2: Yes

5. Is the manuscript presented in an intelligible
fashion and written in standard English?

Reviewer #1: (No Response)

Reviewer #2: Yes

6. Review Comments to the Author

Reviewer #1: (No Response)

Reviewer #2: The authors answered all the issues arisen during the first review. All
the comments have been fully addressed.

7. PLOS authors have the option to publish the peer
review history of their article (what does this mean?). If published, this will
include your full peer review and any attached files.

If you choose “no”, your identity will remain anonymous but your review may still be
made public.

**Do you want your identity to be public for this peer review?** For
information about this choice, including consent withdrawal, please see our
Privacy Policy.

Reviewer #1: No

Reviewer #2: Yes: Nicola Francesco Lopomo

---

## [Editor Report · Acceptance letter]

13 Dec 2019

PONE-D-19-27579R1 

Lateral pressure equalisation as a principle for designing support surfaces to
prevent deep tissue pressure ulcers 

Dear Dr. Boyle:

I am pleased to inform you that your manuscript has been deemed suitable for
publication in PLOS ONE. Congratulations! Your manuscript is now with our production
department. 

With kind regards,

on behalf of

Dr. Pedro H. Oliveira 

Academic Editor

PLOS ONE